# Functional and Biological Characterization of the LGR5Δ5 Splice Variant in HEK293T Cells

**DOI:** 10.3390/ijms252413417

**Published:** 2024-12-14

**Authors:** Matthias Kappler, Laura Thielemann, Markus Glaß, Laura Caggegi, Antje Güttler, Jonas Pyko, Sarah Blauschmidt, Tony Gutschner, Helge Taubert, Sven Otto, Alexander W. Eckert, Frank Tavassol, Matthias Bache, Dirk Vordermark, Tom Kaune, Swetlana Rot

**Affiliations:** 1Department of Oral and Maxillofacial Plastic Surgery, Faculty of Medicine, Martin Luther University Halle-Wittenberg, 06120 Halle, Germanysarah.blauschmidt@uk-halle.de (S.B.); frank.tavassol@uk-halle.de (F.T.);; 2Institute of Molecular Medicine, Section for Molecular Cell Biology, Faculty of Medicine, Martin Luther University Halle-Wittenberg, 06120 Halle, Germany; markus.glass@medizin.uni-halle.de; 3Department of Radiotherapy, Faculty of Medicine, Martin Luther University Halle-Wittenberg, 06120 Halle, Germany; antje.guettler@uk-halle.de (A.G.); matthias.bache@uk-halle.de (M.B.); dirk.vordermark@medizin.uni-halle.de (D.V.); 4Institute of Molecular Medicine, Section for RNA Biology and Pathogenesis, Faculty of Medicine, Martin Luther University Halle-Wittenberg, 06120 Halle, Germany; jonas.weisse@uk-halle.de (J.P.); tony.gutschner@uk-halle.de (T.G.); 5Department of Urology, University Hospital Erlangen, Friedrich-Alexander-University of Erlangen-Nürnberg, 91054 Erlangen, Germany; helge.taubert@uk-erlangen.de; 6Department of Oral and Maxillofacial Surgery and Facial Plastic Surgery, Ludwig Maximilians University, 80337 Munich, Germany; sven.otto@med.lmu.de; 7Department of Cranio Maxillofacial Surgery, Paracelsus Medical University, 90471 Nuremberg, Germany; alexander.eckert@klinikum-nuernberg.de

**Keywords:** LGR5, LGR5Δ5, stem cell marker, isoform, CRISPR/CAS, LGR4

## Abstract

The regulator of the canonical Wnt pathway, leucine-rich repeat-containing G protein-coupled receptor 5 (LGR5), is expressed in the stem cell compartment of several tissues and overexpressed in different human carcinomas. The isoform of the stem cell marker LGR5, named LGR5Δ5 and first described by our group, is associated with prognosis and metastasis in oral squamous cell carcinoma (OSCC) and soft tissue sarcoma (STS). In a proof-of-principle analysis, the function of LGR5Δ5 was investigated in HEK293T cells, a model cell line of the Wnt pathway, compared to full-length LGR5 (FL) expression. The CRISPR/CAS knockout of *LGR5* and *LGR4* (thereby avoiding the side effects of LGR4) resulted in a loss of Wnt activity that cannot be restored by LGR5Δ5 but by LGR5FL rescue. The ability to migrate was not affected by LGR5Δ5, but was reduced by LGR5FL overexpression. The CRISPR/CAS of *LGR4* and *5* induced radiosensitization, which was enhanced by the overexpression of LGR5FL or LGR5Δ5. RNA sequencing analysis revealed a significant increase in the ligand R-spondin 1 (RSPO1) level by LGR5Δ5. Furthermore, LGR5Δ5 appears to be involved in the regulation of genes related to the cytoskeleton, extracellular matrix stiffness, and angiogenesis, while LGR5FL is associated with the regulation of collagens and histone proteins.

## 1. Introduction

The stem cell marker and leucine-rich repeat-containing G protein-coupled receptor LGR5, also known as GPR49, is a positive regulator of Wnt signaling [1]. It was originally identified as a stem cell marker of the small intestine and the colon [2] and has since been found in the stem cell compartment of many tissues [3]. However, LGR5 has also been described as a cancer stem cell marker for various malignancies [4]. Together with their homologs LGR4, these membrane receptors are involved in organ development, stem cell homeostasis, and tumorigenesis. In particular, LGR5 seems to be important for stem cells, whereas LGR4 can be found in early progenitor cells [5,6]. Both receptors (LGR4 and LGR5) are expressed ubiquitously in most organs and tissues, whereas LGR6 is expressed less ubiquitously, e.g., in the stem cells of hair follicles [7,8]. These findings suggest that LGR4 and LGR5 have very similar but also nonoverlapping functions in organ development and stem cell survival, where LGR4 plays a dominant role [9,10]. This is supported by mouse experiments, where the knockout of *LGR5* caused a lethal phenotype, i.e., the fusion of the tongue to the floor of the oral cavity (ankyloglossia) [11]. The developmental lethal defects caused by LGR4 knockout are even worse and disturb different organs [10].

The receptors LGR4 and LGR5 bind members of the secreted ligand R-spondin family (RSPO1-4), which results in the inhibition of two E3 ubiquitin ligases, namely, ring finger protein 43 (RNF43) and zinc and ring finger 3 (ZNRF3) [1,12,13,14]. Recent studies support this mechanism of direct interaction between LGR4/RSPO and E3 ubiquitin ligases, whereas the LGR5/RSPO complex appears to enhance Wnt/β-catenin signaling without directly interacting with E3 ligases [9,15]. RNF43/ZNRF3 usually ubiquitinates Wnt receptors and marks them for degradation. Therefore, *LGR4* and LGR5 activity can protect and increase Wnt receptor levels and thereby enhance Wnt signaling. These findings suggest that LGR4 and LGR5 act as positive regulators of the canonical Wnt signaling pathway [5,16].

Previously, we discovered a splice variant of LGR5 lacking exon 5, which resulted in a truncated RSPO ligand-binding domain called LGR5Δ5 (GPR49Δ5). We described the role of this splice variant in the prognosis and metastatic progression of patients with soft tissue sarcoma [17] and oral squamous cell carcinomas [18]. This isoform has been poorly investigated [19,20]. However, the functional effects of a truncated ligand-binding domain of LGR5 transcript variants have not yet been investigated [18]. “It is unclear how this differential expression of LGR5 isoforms may promote the development of human cancers, particularly with regard to the regulation of Wnt/β-catenin activity” [21].

To address this open question, the Wnt signaling pathway was inhibited by the clustered regularly interspaced short palindromic repeats technique (CRISPR/Cas9) [22] double knockout of *LGR4* and LGR5 in HEK293T cells. This was necessary, as both receptors can take over the function of the other by activating the Wnt pathway. Both full-length LGR5 (wild type) and the splice variant LGR5Δ5 were subsequently reconstituted. The specific effects of LGR5Δ5 in the HEK293T cell line were analyzed in comparison with the effects of LGR5FL rescue.

## 2. Results

### 2.1. Structures of Full-Length LGR5 and LGR5Δ5

In 2010, the sequence of a new splice variant of LGR5, called LGR5Δ5 (GPR49Δ5), was published for the first time by our group and is available in the European Bioinformatics Institute database (EMBL-EBI: https://www.ebi.ac.uk/ena/browser/view/FN820440.1 (accessed on 8 December 2023)).

Compared with the full-length LGR5 mRNA and protein sequences, the skipping of exon 5 results in a truncated variant/isoform, which we named LGR5Δ5 (Figure 1a).

This isoform is shortened by 216 base pairs and 72 amino acids (amino acids 143–214), resulting in a deletion of the leucine-rich region 4–7 (LRR). LRRs 3–9 have been described as an interaction region of LGR5 with the ligand RSPO (Figure 1b).

### 2.2. Internalization of LGR5 Full Length and LGR5Δ5

To study the physiological membranous localization and internalization of overexpressed LGR5FL or LGR5Δ5 isoform, a pulse-chase assay was performed (Figure 2a). These experiments demonstrated that both LGR5 isoforms accumulated in the membrane of HEK293T cells at low temperature (4 °C). Upon transfer to physiological temperature (37 °C), the internalization of both receptors started immediately (5 min), resulting in a significant clearance of the overexpressed receptors within 20 min. Our results showed that the internalization of both LGR5 isoforms was independent of RSPO1, Wnt family member 3A (WNT3a), or their combination. Moreover, the high turnover rate of these receptors made them difficult to visualize on the membrane under physiological conditions (37 °C).

Moreover, to elucidate the mechanism of internalization (endocytosis) of overexpressed LGR5FL or LGR5Δ5 receptors in HEK293T cells, a pulse-chase assay was performed using two inhibitors of a clathrin-dependent internalization (MDC or PitStop) and filipin, an inhibitor of caveolin-mediated endocytosis. The results revealed that the endocytosis of LGR5FL and LGR5Δ5 occurs via clathrin-dependent internalization. (Figure 2b).

These data thus demonstrated a similar physiological behavior of the proteins overexpressed by plasmid constructs, as expected for LGR5 receptor proteins.

### 2.3. Activation of the Wnt Pathway by Full-Length LGR5 and LGR5Δ5

After the validation of the correct membranous localization and the internalization of LGR5FL and LGR5Δ5 proteins translated from artificial constructs, the TOPFlash/FOPFlash reporter assay, a luciferase-linked assay that can quantify Wnt activity in living cells, was used to analyze the functionality of the proteins.

Compared with HEK293T control cells (transfected with an empty vector), LGR5FL-overexpressing cells strongly increased the ability of RSPO1 to amplify Wnt/β-catenin signaling (Figure 3a). These findings prove that the artificial construct for LGR5FL is functionally active. In contrast to our hypothesis, the overexpression of LGR5Δ5 surprisingly also increased Wnt/β-catenin signaling compared to control cells (cells transfected with an empty vector). Compared with these control cells, both LGR5FL- and LGR5Δ5-transfected cells presented a greater basal activity (Figure 3a).

The cell line HEK293T expresses endogenous LGR4, LGR5, and RSPO1 [12], which could contribute to the unexpected result of LGR5Δ5 overexpression. To investigate this potential confounder effect of LGR4, it was knocked down using siRNA treatment (Figure 3b).

This significantly reduced the ability of RSPO1 to amplify Wnt signaling only in vector control cells and LGR5Δ5-expressing cells. (Figure 3b). Notably, the maximum Wnt signaling activity decreased from 70–80 (ratio) without LGR4 knockdown to under 40 after LGR4 knockdown (Figure 3a,b). These data indicate that the endogenous levels of LGR4 dominate these experimental results, particularly for LGR5Δ5-expressing cells. Therefore, to accurately analyze the effect of LGR5Δ5 overexpression, a complete silencing of endogenous LGR4 and LGR5 was required.

### 2.4. Knockout of LGR4 and LGR5 in HEK293T Cells and Rescue of Full-Length LGR5 or LGR5Δ5

To investigate the function of LGR5FL or LGR5Δ5 independent of the effects of endogenous LGR4 and LGR5, both genes were knocked out via CRISPR/Cas9 technology [22] (Figure 4). This was necessary to avoid potential confounding factors and ensure that the observed effects were caused due to the LGR5 isoforms alone. Sanger sequencing identified the specific knockout of *LGR4* and *LGR5* genes, ensuring that the subsequent experiments were realized with cells that lacked endogenous LGR4 and LGR5 expression. The results are shown for a monoclonal double-knockout cell line in comparison to the wild-type sequence in Figure 4 and Appendix A.

In addition, several other single- or double-knockout HEK293T cell lines (knockout of LGR4 or/and LGR5) were generated and used as reference cell lines (see Appendix A). The functional verification of the knockout was realized via a TOPFlash/FOPFlash reporter assay (Figure 5a).

The knockout of endogenous *LGR4* and *LGR5* prevents the activation of the Wnt pathway even after the application of very high levels of exogenous RSPO1 and Wnt3a (Figure 5a,b), which demonstrated the important role of both proteins in Wnt pathway activation. The silencing of the Wnt signal activity could not be restored by the transient overexpression of LGR5Δ5, but by LGR5FL overexpression. This finding highlights the most important difference between LGR5Δ5 and LGR5FL and allows a fast screening of cell lines with a restored Wnt activity after a transient rescue of LGR5FL (Figure 4 and Figure 5b).

Afterward, some of the mutated monoclonal cell lines were lentivirally stably transduced with (a) an empty control vector, (b) a vector expressing LGR5FL, or (c) LGR5Δ5. The relevant RNA-sequence of LGR4/LGR5 of a monoclonal double-knockout cell line transduced with either LGR5FL, the LGR5Δ5 isoform, or an empty control vector is shown in Appendix A. The protein expression of the LGR5Δ5 or LGR5FL was labeled with either a Myc or a YFP (yellow fluorescent protein) tag (Figure 6).

The double-knockout cell line stable transduced with empty control vector or LGR5Δ5 showed complete silencing of the Wnt signal in the TOPFlash/FOPFlash reporter assay. A strong signal was found in LGR5FL rescue cells (Appendix A), very similar to the results of the transient experiment (Figure 5a).

### 2.5. Proliferation and Migration of Full-Length LGR5 and LGR5Δ5 Cells

After stimulation with RSPO1 and Wnt3, no significant changes in the growth behavior of the stable transduced cell lines with LGR5Δ5 or LGR5FL were observed (Appendix A). Only one clone (double-knockout clone 1) with a stable LGR5 rescue showed significantly lower proliferation effects without stimulation (factor of 1.28 ± 0.28 (*p* = 0.04)) compared to the nonmodified control cell line HEK293T (factor of 2.6 (±0.6)) (Appendix A).

Moreover, a wound healing assay was performed to investigate the migration ability of modified LGR4/5 knockout clones (after LGR5FL or LGR5Δ5 rescue) compared with the unmodified cell line HEK293T (Figure 7).

Compared to unmodified HEK293T cells, stable transduction with empty vector or an LGR5Δ5 rescue had no effect on migration. However, the overexpression of LGR5FL resulted in a significant reduction in migration, but only in the double-knockout clones (Figure 7 and Appendix A). No effect of LGR5 modulation on migration was found in a single-knockout cell line (Appendix A), where endogenous LGR4 is expressed.

As a result, it can be assumed that the expression and activation of LGR5FL lead to a reduction in the motility of cells when LGR4 is not present, while the expression of LGR5Δ5 has no such negative effects.

### 2.6. Irradiation of Full-Length LGR5 and LGR5Δ5 Cells

The radiobiological behavior of the two cell lines was compared with that of the unmodified HEK293T cell line. The *LGR4* and *LGR5* double-knockout cell line presented significant radiosensitization, which was further increased after LGR5 modification (Figure 8 and Table 1).

The stable overexpression of LGR5Δ5 led to sensitization to DMF10 of 1.46 ± 0.13 and to LGR5FL of 1.54 ± 0.12 (LGR5) (Table 1 and Figure 8). This effect was not detected in stable LGR5FL-modified single-knockout cells, suggesting that the radiosensitisation is strongly dependent on the absence of LGR4. For this cell line, only LGR5Δ5-overexpressing cells also presented a radiosensitization to DMF10 of 1.17 ± 0.06 (*p* = 0.04) (Appendix A). This effect could be due to the possible negative impact of LGR5Δ5 on the function of LGR4, potentially through the dimerization and neutralization of LGR4 function. In contrast, LGR5FL should not have such an effect on LGR4 function.

### 2.7. Gene Expression Analysis of Full-Length LGR5 and LGR5Δ5 Cells

Two modified HEK293T cell lines (one with a single LGR5 knockout and the other with a double knockout for LGR4/5) and their rescue clones (transduced LGR5Δ5, LGR5FL, or empty vector) were analyzed via RNA deep sequencing experiments. The experiments were performed without stimulation (RSPO1 and Wnt3a for 7 h) to assess the baseline gene expression level. The data of the double-knockout cell lines are presented in Figure 9; the results of the single-knockout clones are shown in the Appendix A as a further reference.

In the LGR4/5 double-knockout cell line, 371 genes were significantly deregulated after LGR5Δ5 overexpression (Appendix A and Figure 9a), whereas only 203 genes were deregulated because of LGR5FL overexpression (Appendix A and Figure 9a). Furthermore, 98 genes were significantly deregulated in both lines (LGR5Δ5 or LGR5FL overexpression) compared with those in the empty vector (PLVX) control cells (Figure 9a).

The most interesting result of this comparison was the expression of RSPO1, the ligand of LGR5 and ‘activator’ of the LGR5/RNF43/Wnt pathway. Compared with control cells, the RSPO1 mRNA level was significantly downregulated in LGR5FL-overexpressing double-knockout cells (log_2_-fold change = −3.71 downregulation). On the other hand, the overexpression of LGR5Δ5 led to a significant 2.3-fold upregulation of RSPO1 mRNA in the double-knockout cell line (Appendix A). This also suggests that LGR5Δ5 may have a diametral effect on RSPO1 expression compared to LGRFL.

In the double-knockout cell line [LGR4/5], only 15 genes (Figure 9a,c) presented opposite deregulation at the mRNA level in response to the overexpression of LGR5Δ5 compared with the overexpression of LGR5FL (Figure 9c), e.g., RSPO1. The expression of LGR5 mRNA was considered as a control and indeed corresponds to either LGR5FL or LGR5Δ5 expression (Appendix A).

Moreover, in the LGR5Δ5-overexpressing cell line (double-knockout [LGR4/5]), the mRNA levels of collagen type XI alpha 1 chain (COL11A1), angiopoietin 1 (ANGPT1), Frizzled-10 (FZD10), telomerase reverse transcriptase (TERT), and xylosyltransferase 1 (XYLT1) (Appendix A and Figure 9b) were also significantly deregulated. COL11A1 (log_2_FC = +4.2) and ANGPT1 (log_2_FC = + 1.2) were strongly upregulated whereas XYLT1 (log_2_FC = 0.7) and FZD10 (log_2_FC = −1.1) were downregulated (Appendix A). A list of genes whose expression was significantly deregulated due to the overexpression of LGR5Δ5 or LGR5FL in different cell lines is shown in Appendix A. KEGG pathway which were enriched because of the overexpression of LGR5FL are shown in Appendix A or after overexpression of LGR5Δ5 (single-knockout [LGR5]) (Appendix A). 

For the pathway analysis of LGR5Δ5-overexpressing cells (371 deregulated genes), 15 ZNF gene products were identified (Appendix A). These ZNF’s comprised 4% of all LGR5Δ5-deregulated gene products, suggesting a specific role in the LGR5Δ5 pathway. The KEGG pathway analysis did not reveal any significantly enriched signaling pathways.

On the other hand, 65 genes were significantly deregulated in both LGR5Δ5-overexpressing cell lines (single- versus double-knockout cells) (Appendix A). The only KEGG pathway found to be significantly enriched was the laminin G pathway (adj. *p* < 0.03 *), containing Col11A1, Col5A1, Gas6, and CELSR5 (Appendix A). The laminin G pathway is involved in the formation of the basal lamina, an important extracellular barrier for the compartmentation of organs. This also suggest that LGR5Δ5 may have a role in the regulation of the basal lamina formation.

Finally, to identify genes activated by stimulation with RSPO1 and Wnt3a, the sequencing data of all, single-knockout, and double-knockout HEK293T cell lines and an unmodified HEK293T cell line were analyzed together (stimulated (with Wnt3a and RSPO1) versus non-stimulated; 7 h) (Appendix A). The timing for this experiment was chosen according to Mahmoudi et al. to identify the target genes regulated by the Wnt pathway in these cell lines [23]. As a result, a list of 27 genes that were deregulated by stimulation with RSPO1 and Wnt3a was subsequently identified (FDR < 0.05; |log_2_(FC)| ≥ 0.5). These genes were identified as potential targets of the Wnt pathway in the HEK293T cell line. Overrepresentation analysis [24] revealed the significant enrichment of Wnt pathway-associated gene products and identified therein 6/27 gene products (Axin2, DKK1, Wnt11, TCF7, LEF, and NKD1) (*p* < 0.001 *) (Appendix A).

### 2.8. Western Blot Analysis

To confirm the RNA sequencing results, several genes deregulated by the LGR5-mediated modulation/activation of the Wnt pathway were tested at the protein level [25]. The translation of RNA into protein obviously requires more time; therefore, proteins were used 48 h post stimulation (as opposed to 7 h for RNA analysis). The effect of the LGR5 modulation (the overexpression of LGR5FL or LGR5Δ5) on level was tested using several genes that were previously identified as deregulated by the LGR5-mediated modulation/activation of the Wnt pathway. This included proteins such as COL11A1, which is deregulated without stimulation, and DKK1, which is only activated after stimulation (Figure 10).

As an example, COL11A1 protein levels were higher in LGR5Δ5-overexpressing cells than in PLVX control- or LGR5FL-expressing double-knockout cell lines, consistent with the RNA data. In the single-knockout cell line, COL11A1 levels were reduced due to LGR5Δ5 rescue compared to PLVX control- or LGR5FL-expressing cells, especially after stimulation. Furthermore, the protein expression of COL11A1 was much greater in the single- knockout (LGR5) cell lines than in the double-knockout cell lines (Figure 10). This suggest that LGR5Δ5 has an impact on COL11A1 in an LGR4-dependent manner.

LGR5FL/LGR5Δ5 was overexpressed in both cell lines (single-knockout (LGR5) and double-knockout HEK293T cell lines), similar to Figure 6, whereas only the construct tagged with a YFP tag (27 kDa) was used for Figure 10. It seems that stimulation had no effect on the expression of the LGR5 protein.

Dickkopf 1 (DKK1), a negative regulator of the Wnt signaling pathway and itself a protein regulated by the Wnt pathway, was upregulated after stimulation compared to the non-stimulated HEK293T cell lines, according to the RNA analysis data, which also confirm that it is a downstream target of the Wnt pathway in HEK293T cell lines. However, the level of the RSPO1 protein was almost unchanged in all the cell lines. 

## 3. Discussion

LGR5 is a well-known stem cell marker that is a regulator of the Wnt pathway and is overexpressed in different human carcinomas (e.g., colon cancer, glioblastoma, and oral squamous cell carcinoma (OSCC)) [18,26,27]. In a recent publication, we reported a significant prognostic association (OS and DSS) and increased risk of metastases of the splice variant LGR5Δ5 [17,18]. However, the molecular and functional basis for these prognostic associations of LGR5Δ5 has not been well investigated. The aim of this study was to identify the molecular, biological, and functional in vitro effects of the overexpression of LGR5Δ5 compared with those of LGR5FL.

The human embryonic kidney cell line HEK293T is a well-established in vitro model for investigating the activity and modulation of the Wnt pathway, especially in the context of LGR4/5 function [1,12], which was therefore used for this approach.

Our initial results confirmed very fast and ligand-independent (RSPO1) internalization of the LGR5FL- and LGR5Δ5-overexpressing proteins in HEK293T cells (Figure 2). Cameron et al. described the RSPO-independent, fast internalization of LGR4 and LGR5 into the intracellular bodies of HEK293 cells [12]. Similarly, our experiments revealed that the LGR5FL and LGR5Δ5 proteins are internalized via a clathrin-dependent mechanism (Figure 2). Clathrin is indeed also required for the internalization of the receptors LGR4/5 [13,14,28]. The internalization and activation of LGR5 isoforms seems to be dependent on the dimerization of LGRs and, moreover, internalization and dimerization experiments were able to prove that hypothesis [13,29,30,31]. Taken together, our results suggest that our constructs (LGR5FL and LGR5Δ5) have biological behavior similar to that reported by other authors regarding LGR4/5.

The well-documented RSPO-dependent activation/rescue of the Wnt pathway by the expression of LGR4, LGR5, or LGR6 [1,12,28] in HEK293T cells was confirmed by the overexpression of LGR5FL but not by the overexpression of LGR5Δ5 (Figure 3a). The high endogenous level of LGR4 seems to dominate Wnt activity in HEK293T cells according to the functional TOPFlash/FOPFlash reporter assay [1]. Therefore, a siRNA-mediated knockdown of LGR4 could clarify whether this protein (LGR4) overshadows the effect of LGR5FL and LGR5Δ5 on the activity of the Wnt pathway. As a result, LGR5FL overexpression was not affected by the endogenous LGR4 level, but LGR5Δ5 overexpression was clearly affected in HEK293T cells (Figure 3b). As a cross-check experiment, the siRNA-mediated silencing of both E3 ubiquitin ligases (RNF43 and ZNRF3) was performed (Appendix A). These findings strongly support the upstream role of RNF43 and ZNRF3 in the activation of the Wnt pathway, which was independent of exogenous RSPO1 levels or LGR5FL overexpression but was supported in an RSPO-dependent manner in LGR5Δ5-overexpressing HEK293T cells (Appendix A). These results demonstrated that the endogenous protein level of LGR4 could modify the effect of LGR5FL but significantly influence the biological effect of LGR5Δ5. This biological interaction between both receptors (LGR4/5) is very likely. LGR4 and LGR5 are homologous receptor proteins that are strongly associated with the Wnt signaling pathway, and the knockout of one of these two genes in mice leads to similar developmental defects and is lethal in each case [11,32,33]. Compared with single knockout, the knockout of both genes (LGR4 and LGR5) exacerbated the malformation of different tissues in mice [13].

To clarify the biological function of LGR5Δ5 or LGR5FL without interference from endogenous LGR4 (or LGR5), the CRISPR/Cas9-mediated knockout of both LGR4 and LGR5 in HEK293T cells was performed. This enables the investigation of markers (such as LGR5Δ5) in a completely silenced genetic background (as clonal cell lines) [22]. Recently, this strategy of developing clonal lines via next-generation CRISPR has become a promising tool for simplifying in vitro disease models [34].

As a result of this strategy, clonal HEK293T cell lines with double (LGR4/5) knockout were produced (Figure 4). Thus, the Wnt activity of those cell lines was strongly reduced as shown in the TOPFlash/FOPFlash reporter assay, but this activity could be restored by the overexpression of LGR5FL in an RSPO1-dependent manner but not by LGR5Δ5 overexpression (Figure 5). This clearly indicates that LGR5Δ5 does not interact with the ligand RSPO1 and is not able to enhance the canonical Wnt pathway without the support of endogenous LGR4 or even endogenous LGR5FL (Figure 1, Figure 3, and Figure 5).

To date, only one study has investigated the biological differences between LGR5FL and its splice variant LGR5Δ5 in colorectal cancer (CRC) and HEK293 cells. Osawa et al. reported significantly decreased Wnt activity (using the TOPFlash/FOPFlash reporter assay) in LGR5FL-overexpressing HEK293 cells compared with that in cells overexpressing splice variants of LGR5 [19]. This does not correspond to our results for the HEK293T cells (Figure 3a). Moreover, the authors speculated that the proliferation rate of LGR5Δ5-expressing cells was increased, whereas LGR5FL was expressed during cell cycle arrest in CRC cells [19]. In our experiments, the proliferation of HEK293T cells overexpressing either LGR5Δ5 or LGR5FL was comparable to that of unmodified control HEK293T cells (Appendix A). However, a significant reduction in the migration ability of the LGR5FL-overexpressing cell line (Figure 7 and Appendix A) was detected when LGR4 was turned off. However, LGR5Δ5 overexpression or empty vector expression caused the same migration as unmodified HEK293T cells (with endogenous LGR4/5 expression). It must be noted, however, that Osawa et al. analyzed the biological differences between natively expressing cells in a completely different cell system (CRC).

Furthermore, the LGR5FL-positive cell population appears to increase after chemotherapy [19] and could therefore indicate therapeutic resistance in the LGR5FL-positive population. The therapeutic test of our approach involved the radiosensitization of such engineered cell lines. However, a significant increase in radiosensitivity was detected in the LGR4 and LGR5 double-knockout cell lines, which was further enhanced by the overexpression of LGR5FL or LGR5Δ5 (Figure 8). The overexpression of LGR5Δ5 or LGR5FL had no clear effects on the radiosensitivity of the cell lines with endogenous LGR4 expression (Appendix A).

Indeed, LGR4 is a regulator of radiosensitivity in cancer, e.g., in prostate cancer cells [35], and the RSPO1–LGR4 axis could be an important protection factor against the radiation effects of mesenchymal bone stem cells [36]. However, LGR5 could also modify radiosensitivity in a sex-dependent manner because LGR5-positive crypt cells are significantly more radioresistant in female mice than in their male counterparts [37].

The proposed therapeutic approach of using LGR5 as a biological target [38] would include both proteins, LGR5FL and LGR5Δ5. However, the specific function and associated genes regulated by LGR5Δ5 have not been studied. Therefore, it may be difficult to properly assess the consequences of a potential LGR5-targeting therapy on LGR5Δ5 and LGR5Δ5 target genes.

Therefore, the underlying molecular mechanisms and signaling pathways associated with the empiric results of LGF5FL/LGR5Δ5-overexpressing cells could be elucidated via RNA sequencing analysis. These findings provide important information for possible future LGR5Δ5-specific therapeutics.

Although microarray platforms are often used to analyze Wnt/LGR5-associated gene expression [20,23,25], deep sequencing technology has some advantages [39], e.g., isoform-specific analysis.

An important question, which is clarified by sequencing, is which genes are Wnt target genes and are deregulated by stimulation. However, a “large majority were unique to the HEK293 kidney cells” [23]. Following [23], we analyzed our LGR5Δ5/LGF5FL-modified HEK293T cells 7 h after stimulation. The combination of results for multiple cell lines yielded a list of 27 genes (e.g., Axin2, TCF7, LEF1, DKK1 Wnt11, and SP5) (Appendix A). In total, 3/27 genes (AXIN2, TCF7, and Wnt11) were explicitly described as Wnt target genes in HEK293T cells [23]. Moreover, 4/27 genes (Axin2, TCF7, SP5, and PHLDB2) were identified by Munoz et al. and listed as intestinal stem cell signatures of LGR5-positive stem cell-enriched genes (Appendix A, published by [25]).

Another experimental approach was to identify genes that were activated/deregulated in HEK293T cells due to the overexpression of LGR5Δ5 or LGR5FL independent of the activation of the canonical Wnt pathway because of a lack of stimulation (Figure 9 and Appendix A) (e.g., LCP1, Col11A1, Col5A1, FGF17, and RAB38). These five genes were also identified by [25]. This strongly suggests that not all “Wnt pathway genes” are targets of an active canonical Wnt pathway but are more likely LGR5-regulated genes of a noncanonical Wnt pathway or even an LGR5-specific pathway. Nevertheless, the network between the genes associated with canonical or noncanonical Wnt pathways seems to be very close for many diseases, e.g., lipid accumulation, fibrosis, diabetes, cancer, and chronic low-grade inflammation [40,41].

The most interesting and significantly deregulated mRNA was R-spondin 1 (RSPO1), the ligand of LGR5, which was downregulated in the LGR5FL rescue cell lines and upregulated in the LGR5Δ5 rescue double-knockout cell lines (Figure 9). R-spondins regulate, e.g., matrix components (extracellular proteins) in mouse organoid cultures, e.g., collagens such as COL4A1 and COL4A2 [42]. However, in our experiments, COL4A1 (FDR < 0.01) and COL4A2 (FDR < 0.001) were significantly downregulated, whereas COL11A1 was strongly upregulated due to LGR5Δ5 overexpression (Figure 9 and Appendix A). This influence was extended to other collagens in LGR5FL-overexpressing cells (Appendix A). COL11A1 was found to be a prognostic factor in OSCC [43] that promoted the conversion of cancer-associated fibroblasts during pancreatic cancer progression [44] and seemed to promote the migration and invasion of lung adenocarcinoma cells [45]. COL11A1 (and COL22A1) was also more highly expressed in LGR5+ chondrocytes than in LGR5- cells [46]. Therefore, the link between the deregulation of collagens, e.g., Col11A1, and an LGR5Δ5 overexpression (Figure 9 and Figure 10) could be a possible explanation for the described associations of LGR5Δ5 level with metastasis and prognosis in OSCC patients [18]. This LGR5-RSPO–collagen axis could also be important for understanding the ability of some cell lines to form organoids because of the overexpression of LGR5Fl or LGR5Δ5 [20] (Appendix A). Our findings of LGR5/LGR5Δ5-dependent collagen regulation are supported by other results suggesting a role for LGR5 in mammary gland organogenesis [47,48]. Therefore, we suggest that the regulation of the collagen network by LGR5/LGR5Δ5 could be part of ECM remodeling in tumors (reviewed in [49]). These authors explicitly described the role of COL4 and COL11A1 in matrix stiffness and tumor progression.

The most downregulated mRNA in both the LGR5FL and LGR5Δ5 rescue cell lines was lymphocyte cytosolic protein 1/L-plastin (LCP1), a cytoskeleton-related protein that is an actin-binding protein. This gene was also found in LGR5-associated intestinal stem cells [25]. On the other hand, one of the most upregulated mRNAs in both the LGR5FL and LGR5Δ5 rescue cell lines (double-knockout cells) in our experiments was integral membrane protein 2A (ITM2A), which is expressed in human brain endothelial cells and human microvessels and also in HEK293 cells [50]. The ITM2A protein seems to be downregulated when endothelial cells are grown in culture; therefore, Cegarra et al. postulated a role for this protein in brain transcytosis or transport [50]. A second secreted glycoprotein that plays a role in vascular development and angiogenesis is angiopontin (ANGPT1), which was significantly upregulated only in the LGR5Δ5 rescue cell lines of both the double- and single-knockout cell lines (Figure 9 and Appendix A). These data support the known role of the Wnt pathway in angiogenesis [51], which has already been postulated even for LGR5 [52]. Our data suggest that LGR5Δ5 could promote angiogenesis via the activation of ANGPT1 and ITM2A. Munoz et al. reported that the analog gene (ANGPT2) is regulated by LGR5 in mice [25]. Because angiogenesis is especially connected with metastasis and tumor progression, this link could be a second possible explanation for the described associations of LGR5Δ5 with metastasis and prognosis in OSCC patients [18], in addition to the postulated function of LGR5Δ5 in the stiffness of the ECM (Figure 11).

In our study, hTERT was significantly downregulated in HEK293T cells overexpressing LGR5Δ5 (Appendix A, Figure 9); LGR5FL-overexpressing cells presented the same expression pattern as control cells. However, human telomerase activity is increased in primary isolated LGR5-positive cells, but these cells present stem cell characteristics that are not present in our HEK293T cells [53].

In general, LGR5Δ5 seems to be strikingly higher expressed (>30%) even in ‘normal’ unmodified HEK293T cell, if all LGR5 isoforms are higher expressed (Appendix A). This increased expression of LGR5 isoforms was independent of a stimulation with WNT3a and RSPO1. These data also suggested that an unknown stimulus can activate LGR5 expression and because of such a higher expression, a splicing process could generate even higher levels of LGR5Δ5. It is not known if the expression of the isoforms occurs in the same cells or in different cells; for such an investigation, an LGR5Δ5-specific antibody is necessary. It is not unlikely that even LGR5∆5 is involved as a homo- or heterodimer partner in the canonical Wnt signaling pathway and may also be involved in other signal transduction pathways that are independent of RSPO action [20]. Thus, LGR5FL was shown to activate the G12/13 Rho GTPase signaling pathway, affecting NF-κB and c-fos, targets of the Rho GTPase, which was independent of R-spondins [54]. This function could therefore also be performed by LGR5∆5.

For some years, the important role of the LGR5- R-spondin pathway in craniofacial development has been described [55,56,57,58], and the findings support the role of LGR5 in the head and neck region. The therapeutic approach of using LGR5 even as a biological target [38] was recently very successful in patients with recurrent or metastatic head and neck squamous cell carcinoma (HNSCC) as a phase 2 study (NCT03526835) [59]. Our results concerning the prognostic importance of the splice variant of LGR5 (LGR5Δ5) for specific head and neck cancer (OSCC) patients require two open questions. The role of LGR5Δ5 in OSCC cells is unknown, and LGR5Δ5 (e.g., LGR5Δ5-specific therapeutic antibody) may be an even better therapeutic target than LGR5FL-specific antibodies. Therefore, LGR5Δ5 should be investigated in in vitro HNSCC (OSCC) models to clarify these open questions.

## 4. Materials and Methods

### 4.1. Cell Cultivation

The HEK293T cell line (ATCC CRL-3216) was cultivated at 37 °C and 5% CO_2_ in DMEM (PAA Laboratories, Pasching, Austria or Sigma-Aldrich, Taufkirchen, Germany) supplemented with 10% fetal calf serum (FCS) (Capricorn Scientific, Ebsdorfergrund, Germany) and 2% antibiotics (185 U/mL penicillin, 185 µg/mL streptomycin (Invitrogen, Darmstadt, Germany)) as adherent monolayers. To perform the TOPFlash/FOPFlash reporter assay, the cells were solubilized with Accutase (PAA Laboratories, Pasching, Austria or Biowest, Nuaillé, France). The desired number of cells was seeded in 96-well plates (between 5000 and 25,000 cells/well). To produce spheroid cultures (3D culture), 96-well plates were coated with 1.5% agarose in DMEM. Subsequently, 3000 to 7000 cells/well were streaked and cultured for four to seven days. The morphology of the spheroids was assessed and documented under a microscope. Tests for mycoplasma contamination were carried out regularly.

### 4.2. Lentiviral Transduction

HEK293T and CRISPR clones (sequence of the used gRNA see Appendix A) of the HEK293T cell line [22] were used, in which either only LGR5 or the homologs LGR4 and LGR5 had been mutated and verified in advance as mutated at the RNA and DNA levels. The LGR5 and LGR5Δ5 receptors were restored in these CRISPR clones with the help of lentiviruses. The HEK293T cell line was selected as the producer cell line for the production of the lentiviral particles. For this purpose, 2 × 10^6^ cells were seeded in a 10 cm Petri dish, and the expression plasmid pLVX (10 µg) (Addgene (#12260)) and the packaging plasmids pMD2. G (5 µg) (Addgene (#12259)) and psPAX2 (10 µg) (Addgene (#12260) (Addgene, Watertown, MA 02472, USA) were co-transfected the following day via the calcium phosphate method [20]. For this purpose, the corresponding amounts of expression and packaging plasmids were mixed in a reaction vessel, with 0.5 mL of sterile buffered water, 0.5 mL of 2× HBS, and 60 µL of 2.5 M CaCl_2_ solution added dropwise per transfection batch [20]. After an incubation time of 20 min at room temperature, 1 mL of the turbid transfection mixture was added to the cell medium (10 mL) (start of S2 safety work). Different Petri dishes of HEK293T cells were treated with different transfection solutions, which always contained the two packaging plasmids and either the LGR5, LGR5Δ5, or the pLKO.1 empty vector (control) to be transfected. Sixteen hours after transfection, the medium was changed by adding 10 mL of fresh complete DMEM, and the virus supernatant was harvested 48 h after transfection. The cell mixture was carefully removed and sterile-filtered through a 0.4 µm filter to remove floating cells and cell debris. The virus-containing supernatants were stored in a refrigerator at 4 °C and could be added to the different CRISPR clones or nonmodified cell lines in the following step, whereby all the cell line clones were treated with 3 different constructs (transgene LGR5, LGR5Δ5, or empty vector). For infection, 2 × 10^6^ cells of each cell clone were seeded three times in a 6-well plate, cultivated overnight, and infected the following day with 300 µL of the respective lentiviruses per well. Selection was performed continuously by adding 2 µg/mL puromycin to the cell culture medium for the first time 48 h after infection (puromycin resistance of the transfected cells). After 5 passages, the cells were transferred to the S1 range. By integrating a YFP tag into the LGR5 and LGR5Δ5 expression plasmids, the cells were sorted every 4 weeks on a FACS Aria ll flow cytometer for positively transduced cells to guarantee the enrichment of at least 60% positive LGR5 or LGR5Δ5 cells in the experiments. The cell clones transduced with the Myc tag could not be enriched. In addition, all the cell lines transduced with lentiviruses were cultured with 2 µg/mL puromycin in DMEM.

### 4.3. Pulse-Chase Assay

To investigate the internalization of the overexpressed LGR5FL and LGR5Δ5 proteins, HEK293T cells were cultured on polylysine-coated coverslips. Prior to this, the cells were incubated for 16 h with agonists of the Wnt signaling pathway (150 ng/mL recombinant human Wnt3A and/or 150 ng/mL recombinant human RSPO1 (R&D Systems, Minneapolis, MN, USA)) or for 30 min with endocytosis inhibitors; the inhibition of clathrin-mediated endocytosis was performed with 40 µM MDC (monodansylcadaverine) (Sigma, Steinheim, Germany) and 0.03 µM Pitstop2™ (Abcam, Cambridge, UK), and the inhibition of caveolin-mediated endocytosis was performed with 10 µM filipin III (Sigma, Steinheim, Germany). The vital cells were then incubated with a FITC-conjugated anti-Myc tag antibody (Myc tag clone 9B11; 1:200 cell signal), which is directed against the N-terminal Myc tag coupled to the overexpressed LGR5FL or LGR5Δ5, for 2 h at 4 °C. The cells were then fixed immediately for 5 or 20 min at 37 °C in culture medium without (0% FCS) or with the addition of stimulants before fixation. The cells were fixed by adding 4% PFA (paraformadehyde) in PBS (Sigma Aldrich) for 10 min at room temperature [20]. The cell nuclei were counterstained with DAPI solution (blue).

### 4.4. TOPFlash/FOPFlash Reporter Assay

To quantify the intracellular activity of the canonical Wnt/β-catenin signaling pathway [12] and to functionally confirm the restoration of the receptors, the TOPFlash/FOPFlash reporter assay, a luciferase promoter gene assay, was performed. The transcription factor TCF/LEF is activated by interaction with the coactivator β-catenin and binds specifically to the seven native TCF/LEF binding sites (TOPFlash M50, Addgene (#12456)) or less well to the six mutated TCF/LEF binding sites (FOPFlash M51, Addgene (#12457) (Addgene, Watertown, MA 02472, USA) as a background match). As an internal standard, renilla luciferase (from renilla reniformis, rLuc (pGL4.73 [hRluc/SV40] Promega, Mannheim, Germany)) with different substrate specificities was co-transfected under the control of a constitutive promoter and measured after firefly luciferase quantification. A decrease in renilla luminescence indicates insufficient transfectability. For the assay, 5 × 104 cells/well were seeded in poly-D-lysine-coated 96-well plates and co-transfected after 8 h with the vectors M50-TOPFlash or M51-FOPFlash and the renilla luciferase vector pGL4.73 and incubated overnight at 37 °C in an incubator. For transient transfection, 0.8 µL of transfection reagent/well (Viafect, Promega, Mannheim, Germany) with 200 ng of vector DNA was incubated in serum-free culture medium (Σ10 µL) for 15 min at room temperature, and the transfection mixture was added to 90 µL of the cell suspension.

After 24 h, the medium containing the transfection mixture was replaced with 50 µL of DMEM without phenol red (D1145) supplemented with serum (0.1–1%), 0–200 ng/mL RSPO1, and 25 ng/mL Wnt3a (Figure 5a) or 10 ng/mL Wnt3a (Figure 3a), and the cells were stimulated. After 16 h, luminescence was measured via the Dual-Glo^®^ Luciferase Assay System (Promega, Mannheim, Germany) according to the manufacturer’s instructions. For this purpose, 50 µL of Dual-Glo^®^ luciferase reagent was pipetted onto an equivalent amount of medium that induces cell lysis and serves as a substrate for firefly luciferase. After 10 min of light-protected incubation at room temperature, the firefly luciferase activity was visualized with a ChemiDocTM Touch Imaging System, and the luminescence was quantified with a Spark plate reader or a GENios reader (Tecan reader, Männedorf, Swiss). By adding another 50 µL of Dual-Glo^®^ Stop&Glo^®^ reagent, the luminescence of the firefly luciferase was stopped, and the substrate for the renilla luciferase was provided and measured analogously in the plate reader. Measurements were performed in triplicate, and the experiment was performed in three biological replicates. (Fit: slogistic3 function using (OriginLab Corporation Origin 2019) y(x) = a/(1 + b × exp(−k × x)).

If genes were inhibited prior to the experiments, specific siRNA constructs were used (Silencer select val siRNA RNF43 (ID s29698), LGR4 (ID s229316), and ZNRF3 (ID s38543)) (Thermo Fisher, Waltham, MA, USA). For this reaction, the seeded cells were incubated for 24 h with a 10 nM solution containing the infection agent INTERFERin^®^ transfection reagent (Polyplus Illkirch, France) according to the protocols of the producer and then treated as described above. The transient overexpression of LGR5FL (Carmon 2011) or LGR5Δ5 was performed via pIRESpuro3 transfection [20].

### 4.5. Proliferation Analysis (MTS Assay)

To analyze the influence of LGR4 and LGR5 knockout or the overexpression of LGR5FL or LGR5Δ5 on in vitro cell growth, an MTS assay was performed with all the cell line clones. For this purpose, 25 × 10^3^ cells/well of each cell line were seeded in 200 µL of complete DMEM in a poly-D-lysine-coated 96-well plate. After 24 h, 48 h, and 72 h of incubation (37 °C/5% CO_2_), the MTS assay was performed according to the manufacturer’s instructions (Abcam, Cambridge, UK) by replacing the old cell medium with new medium, adding 20 µL of ready-to-use MTS reagent (1:10) per well, and incubating the cells for 3 h at 37 °C in a cell line incubator. Two experimental conditions were chosen for this purpose: (I) growth under standard conditions with 10% FCS and DMEM (Appendix A) and (II) growth corresponding to the conditions of the migration experiment with DMEM, 1% FCS, 25 ng/mL Wnt3a, and 100 ng/mL RSPO1 (Appendix A). The growth rates were determined over a period of 24 h in one LGR5 knockout cell line and two LGR4/LGR5 double-knockout cell lines with stable LGR5 (FL) or LGR5Δ5 (D5). The amount of color product formed is proportional to the number of cells and the duration of incubation with the MTS/PMS reagent, allowing the number of vital cells to be quantified colorimetrically at different time points. After incubation, the plate was shaken briefly at room temperature, and the absorbance was measured at 490 nm on a Spark plate reader (Tecan, Männedorf, Swiss). The average multiplication of the cell numbers over 24 h could be calculated from the absorbances of the 3 measurement days. Plot of the relative growth rates measured in the MTS assay (quotient MTS absorbance value 48 h/MTS absorbance value 24 h) with LGR5 (FL = blue) or LGR5Δ5 (D5 = red) overexpression compared with the empty vector control (empty = green) and nonmodified HEK293T (control = black) cells. n = 3, *p* value: two-sided, paired t test with 25 ng/mL Wnt3a and 100 ng/mL RSPO1 and 1% FCS or Appendix A without Wnt3a or RSPO1 but supplemented with 10% FCS.

### 4.6. Migration Analysis (Scratch Assay)

The scratch assay allows the analysis of the cell migration rate with respect to receptor modulation, LGR5/LGR4 knockout, and LGR5 or LGR5Δ5 overexpression. For this purpose, 5 × 10^4^ cells/well of each cell line were seeded into a 96-well poly-D-lysine-coated ImageLock plate (Biosciences, Essen, Germany) and incubated for 24 h in an incubator (37 °C/5% CO_2_). The cells (approx. 90–100% confluence) were subsequently incubated for 3 h with 5 µg/mL of the cytostatic drug mitomycin C, which blocks the replication of the cells. This treatment enables the targeted assessment of the migration rate, independent of the effects on cell proliferation. The concentration and incubation time of mitomycin C were determined in advance. After the incubation period, the medium was removed from the cells and replaced with 100 µL of fresh DMEM. To analyze the influence of the Wnt/β-catenin signaling pathway and ensure the activation of the LGR5 receptors, the stimulants Wnt3a (25 ng/mL) and RSPO1 (100 ng/mL) were added to the cell culture medium supplemented with serum (1% serum). As a control, cells with only 1% serum, without stimulants, were included. After adding the medium with and without stimulants, a defined wound 700–800 μm in size was created in the confluent monolayer via the IncuCyte^®^ Wound-MakerTM 96-well tool (Biosciences, Essen, Germany). The migration rate of the cells into the wound could be documented microscopically on the IncuCyte^®^, whereby two phase-contrast images per well were taken every 4 h over a 24 h interval. The relative wound confluence with respect to time 0 was subsequently quantified via scratch wound analysis software by selecting a value of 1.1 for segmentation (background vs. cells), a hole size of 1 × 10^4^ µm^2^, and a minimum cell size of 5000 µm^2^. This allows the system to analyze only the desired wound for migration and does not detect any cell-free areas in the cell lawn. For each biological experiment (N = 3), the mean values were calculated from technical triplicates per cell line and per treatment. *p* value: two-tailed, paired t test for empty vector controls. (Fit: expon. growth 1 function using (OriginLab Corporation Origin 2019) y(x) = a1 × exp(x/t) + yo).

### 4.7. Cell Colony Formation and Irradiation

The cell colony formation test is an endpoint measurement that can determine the clonogenic long-term survival of cell lines and the number of cells still capable of dividing after treatment. LGR4 and LGR5 knockouts as well as LGR5- or LGR5Δ5-modulated cell clones were used, and their radiobiological behavior was analyzed. This involves determining the number of cells that are still able to divide and form macroscopically visible colonies after irradiation (>50 cells). This allows the plating efficiency of each treatment to be calculated. For each cell line and irradiation dose, the optimum cell number was determined in preliminary tests to ensure that the number of colonies formed was within measurable ranges (>20 colonies/bottle).

For the analysis, the defined cell numbers were seeded in 25 cm^2^ cell culture flasks without a filter (closed lid) in 3 mL of complete RPMI (not DMEM) medium in technical duplicates. After 6 h, the flasks were irradiated with 2, 4, 6, or 8 Gy of the linear accelerator Synergy^®^ from Elekta (dose rate of 2 Gy/min with 6 MV photons) in the Department of Radiotherapy of the University Hospital Halle (Saale). The cell culture flasks were then incubated at 37 °C in an incubator, and after 10–14 days, they were fixed with a 3.7% formalin solution, incubated for 20 min, and dried overnight. On the next day, the cell colonies were stained with 5 mL of 20% Giemsa solution, rinsed with distilled water, and air-dried for 2 days. Using GelCount™ (Oxford Optronix, Adderbury, UK), the number of cell colonies per bottle was determined, and the relative plating efficiencies were calculated and plotted semi-logarithmically in dose-survival curves as a function of the irradiation dose.

The linear-quadratic model in Origin software (y(x)= c × exp(−(ax + bx^2^)) was used to fit the curve and determine the dose-modifying factor-10 (DMF10). DMF10 indicates how strongly a treatment (LGR5 modulation) influences radiosensitivity to reduce the relative plating efficiency of a cell line to 10% compared with that of the untreated control. The quotient of the dose at which 10% of the cells still survived was formed from the untreated cell line HEK293T by the modulated HEK clones. A DMF > 1 thus indicates radiosensitization, whereas a DMF < 1 indicates radioresistance. A DMF10 of 1 corresponds to the unmodified control (HEK293T), whereby no effect of irradiation occurred.

### 4.8. Western Blot

For protein analysis, 5 × 10^5^ cells/well (6-well plate) for each cell line were seeded under reduced-calf serum conditions (1% FCS) with or without stimulation (1% FCS, 25 ng/mL Wnt3a, and 100 ng/mL RSPO1) in 2 mL of DMEM and harvested after 48 h of cultivation. For this purpose, the medium was removed, and each cell line was removed in 100 µL of ice-cold lysis buffer supplemented with protease inhibitors. Western blot analysis was performed as described in a previous study [39]. The membranes were subsequently incubated at 4 °C overnight in TBS (50 mM NaCl, 30 mM Tris-HCl pH 7.5) with 2% BSA (Carl Roth, Karlsruhe, Germany) with the primary anti-β-actin monoclonal antibody (mAb) (Clone AC-15, dilution 1:3000, host mouse) from Sigma (Steinheim, Germany), the anti-LGR5 mAb (LGR5 antibody (GTX62071) (EPR3065Y), dilution 1:2000, host rabbit) from GeneTex, Inc. (Alton Pkwy Irvine, CA, USA), the anti-Dkk-1-antibody (sc-374574 (clone B-7), dilution 1:500, host mouse) from Santa Cruz Biotechnology, Inc. (Dallas, TX, USA), the anti-R-spondin 1-antibody (422407 R&D Systems, Minneapolis, MN, USA. dilution 1:500, host mouse), or the anti-COL11A1 antibody (clone (E6O7R), dilution 1:2000, host rabbit) from Cell Signaling Technology (Leiden, The Netherlands).

### 4.9. RNA Isolation and Deep Sequencing Analysis

For the isolation of RNA samples for deep sequencing analysis and for quantification via real-time PCR, 4 × 10^5^ cells per well (12-well plate) of each cell line clone and the control cell line HEK293T were cultivated in 1 mL of complete DMEM overnight at 37 °C/5% CO_2_ to form a confluent monolayer. After 24 h, the medium of each cell line was replaced with non-stimulated, serum-reduced medium (1% FCS) or stimulated medium (1% FCS, 25 ng/mL Wnt3a, and 100 ng/mL RSPO1) to obtain samples from each cell line with an activated and inactive Wnt/β-catenin signaling pathway. Stimulation was performed for 7 h according to [23] and previous control experiments before the RNA was taken up in 300 µL of lysis buffer and isolated via Zymo-Spin™ IIIC/IIC columns with RNA Prep and RNA Wash buffer according to the manufacturer’s instructions via the RNA MiniprepTM Kit (Zymo Research). For each sample, isolation was repeated on three different days (n = 3). Finally, the purified RNA was eluted from the column in 50 µL of RNase-free water and stored at −80 °C until further use. The RNA concentration of each sample and the RNA quality were checked via a NanoDrop 2000c spectrophotometer (Thermo Fisher Scientific (Darmstadt, Germany)). For deep sequencing analysis (genewiz, Leipzig, Germany), 2 µg of each sample was aliquoted and stored on dry ice.

Library synthesis and strand-specific paired-end sequencing (2 × 150 bp) of poly-A-enriched RNA were performed by GENEWIZ (Leipzig, Germany) on an Illumina NovaSeq 6000 device. On average, 2 × 25 million reads/sample were sequenced.

For RNA-seq data analyses, adapter sequences as well as low-quality read ends were clipped off via Cutadapt (v 2.8) [60]. The processed sequencing reads were aligned to the human reference genome (UCSC hg38) via HISAT2 (v 2.1.0) [61]. SAMtools (v 1.10) [62] was used to extract primary alignments and to index the resulting bam files. FeatureCounts (v 2.0.0) [63] was used to summarize the gene-mapped reads. ENSEMBL (GRCmh38 v100) [64] was used as an annotation basis. Differential gene expression was determined via the R package edgeR (v 3.38.1) via trimmed mean of M values (TMM) normalization [65,66]. EdgeR’s exactTest(v 3.38.1) function was used to assess differentially expressed genes. A false discovery rate-adjusted *p* value (Benjamini-Hochberg) below 0.05 was considered the threshold for the determination of differential gene expression. For overrepresentation analysis, enriched genes were used in the program david data (https://david.ncifcrf.gov/ (accessed on 2022-2024)) [24]. Genes were defined as deregulated when the FDR ≤ 0.05 and |log_2_(FC)| ≥ 0.5.

### 4.10. Statistical Analysis

All experiments in this work were carried out in at least three biological replicates. The significance of the data was determined with Microsoft Excel via two-sided, unpaired Student’s *t* test; values ≤ 0.05 were considered significant (*), and *p* values ≤ 0.01 were considered highly significant (**). *p* values for ORA were Benjamini-Hochberg-adjusted.

## 5. Conclusions

Our data suggest that LGR5Δ5 overexpression is unable to rescue the canonical Wnt pathway without support from LGR4 or LGR5FL. However, this isoform appears to be involved in the activation of different genes, partly distinct from the effect of LGR5FL, such as the ligand of the LGR5 pathway, RSPO. LGR5Δ5 was able to sensitize cells to irradiation, likely due to an effect on LGR4, and influenced angiogenesis and the collagen network-associated genes. This study analyzed the effects of LGR5Δ5 and LGR5FL in HEK293T cells, and it would be interesting to expand this initial experiment to other cell lines.

## Figures and Tables

**Figure 1 ijms-25-13417-f001:**
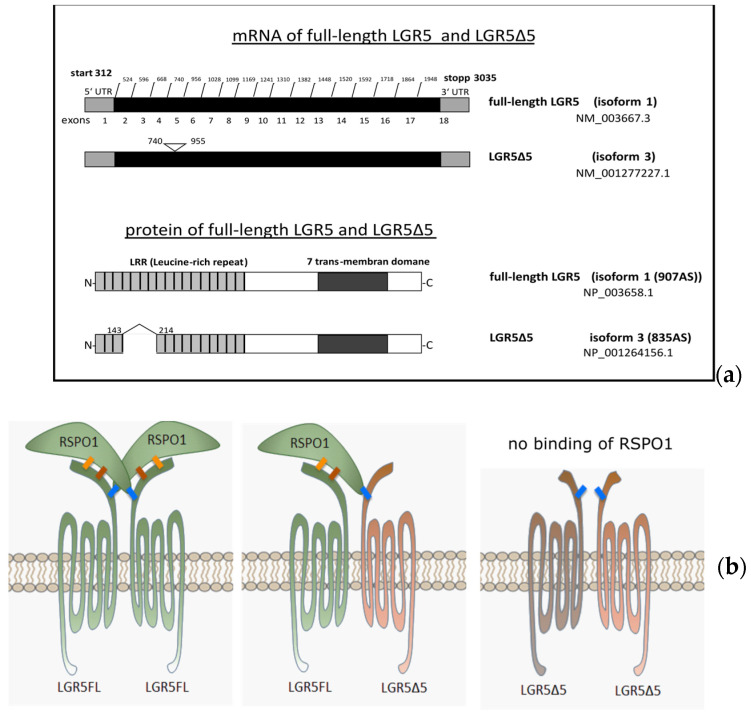
(**a**) Structures of full-length LGR5 and LGR5Δ5 splice variants. The triangles mark the spliced-out site in the mRNA. The corresponding potential proteins are labeled with the NCBI conserved domain search tool. The brackets indicate the spliced-out site in the protein [20]. The substrate binding site was predicted via comparison via NCBI conserved domains. https://www.ncbi.nlm.nih.gov/Structure/cdd/wrpsb.cgi (accessed on 8 February 2016). (**b**) Schematic representation of the possible interaction between RSPO1 and LGR5FL and LGR5Δ5. LGR5FL forms homodimers and is also capable of forming heterodimers with the splice variant LGR5Δ5. RSPO interacts with LGR5FL at three contact sites (1, 2, and 3 in blue, brown, and yellow, respectively). In the LGR5 splice variant Δ5, contact sites 1 and 2 are spliced out. Interaction with RSPO at contact site 3 is possible via heterodimerization with LGR5FL [20].

**Figure 2 ijms-25-13417-f002:**
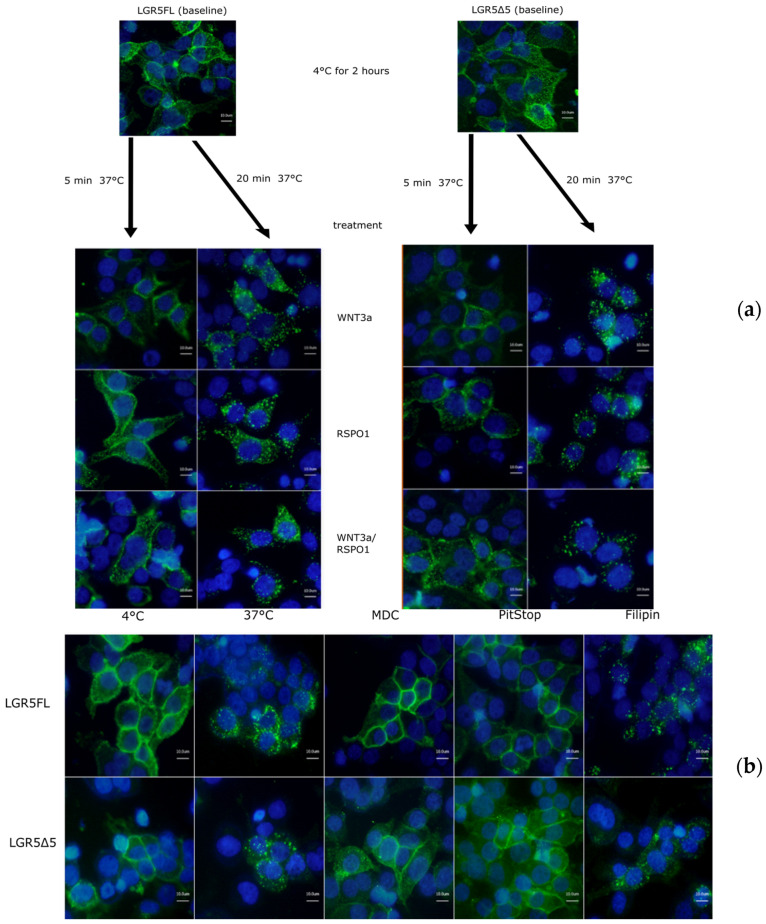
(**a**) Time- and stimulant-dependent internalization of overexpressed LGR5FL and LGR5Δ5 in HEK293T cells. The samples were stained with a FITC-coupled anti-Myc-Tag antibody (green) and counterstained (nuclei) with DAPI solution (blue). (**b**) Inhibition of clathrin- and caveolin-mediated endocytosis in LGR5FL-overexpressing and LGR5Δ5-overexpressing HEK293T cells. The cells were treated with inhibitors of clathrin-mediated endocytosis (MDC and PitStop2) and with an inhibitor of caveolin-mediated endocytosis (filipin III). (The scale bar represent 10µm). [20].

**Figure 3 ijms-25-13417-f003:**
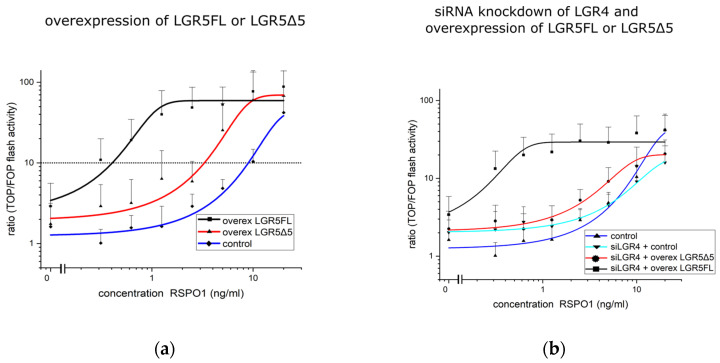
TOPFlash/FOPFlash reporter assay to measure Wnt pathway activity in modified HEK293T cells (**a**) overexpressing an empty vector (blue line), LGR5FL (black line), or LGR5Δ5 (red line). (**b**) siRNA-mediated knockdown of LGR4 and overexpression of an empty vector (light blue line), LGR5FL (black line), or LGR5Δ5 (red line). The control (dark blue line) corresponds to the empty vector control of (**a**).

**Figure 4 ijms-25-13417-f004:**
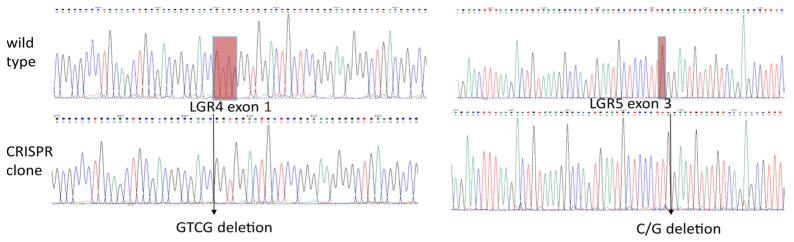
Sanger sequencing of one CRISPR/Cas9 monoclonal cell line of HEK293T cells compared with the wild-type sequence. LGR4 exon 1 (−4 nucleotides) resulted in a stop codon in 629 and a 141 base pair mRNA encoding a truncated protein of 46 As), and LGR5 (−1 nucleotide) resulted in a stop codon in 661 and a 378 base pair mRNA encoding a truncated protein of 125 As. (Please also see the sequence of the rescue clones in Appendix A).

**Figure 5 ijms-25-13417-f005:**
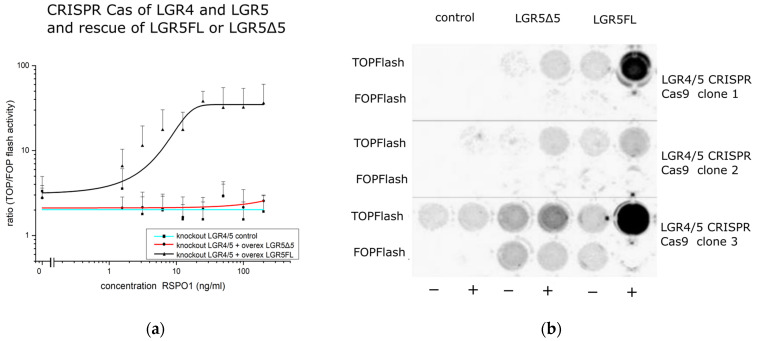
(**a**) TOP/FOP flash assay to measure Wnt pathway activity in modified HEK293T cells (empty vector (blue line), LGR5FL (black line), or LGR5Δ5 (red line) overexpression). (**b**) Screening of selected LGR4/5 CRISPR double-knockout clones via the TOPFlash/FOPFlash reporter assay after LGR5FL or LGRΔ5 rescue. Analysis of the chemiluminescence signals (luciferase activity) from transiently (pIRESpuro) LGR5FL- or LGR5Δ5-transfected cells with (+) and without (−) stimulation with 100 ng/mL Wnt3a and 100 ng/mL RSPO1. Control = knockout clones without transfection. Single-screen transfection was performed via ViaFect™. Clonal cell lines 1 and 3 were validated by sequencing analysis, whereas clone 2 was excluded.

**Figure 6 ijms-25-13417-f006:**
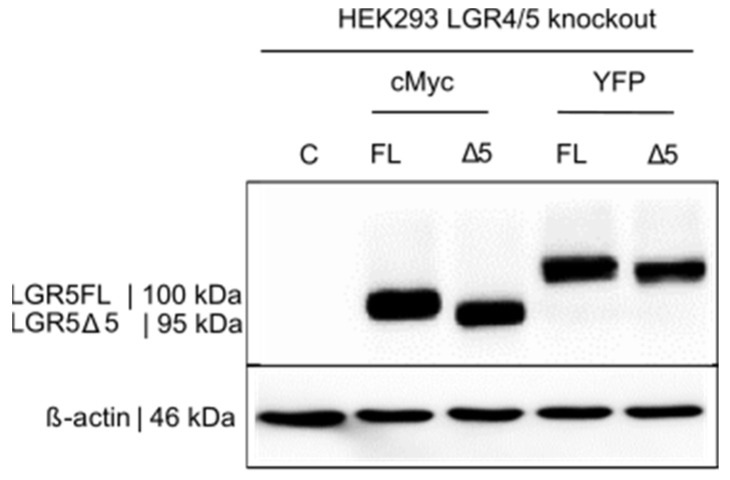
Western blot analysis to detect full-length LGR5 (FL) and LGR5Δ5 (Δ5) expression via the lentiviral transduction of HEK LGR4/5 CRISPR/Cas9 double-knockout cells in comparison with the empty vector control (C). Full-length LGR5 (FL) and LGR5Δ5 (Δ5) were each tagged with a c-Myc tag (<1 kDa) or a YFP tag (27 kDa) for selection, visualization, and enrichment. The detection of LGR5 and LGR5Δ5 was carried out via a polyclonal anti-LGR5 antibody. ß actin was used as a loading control.

**Figure 7 ijms-25-13417-f007:**
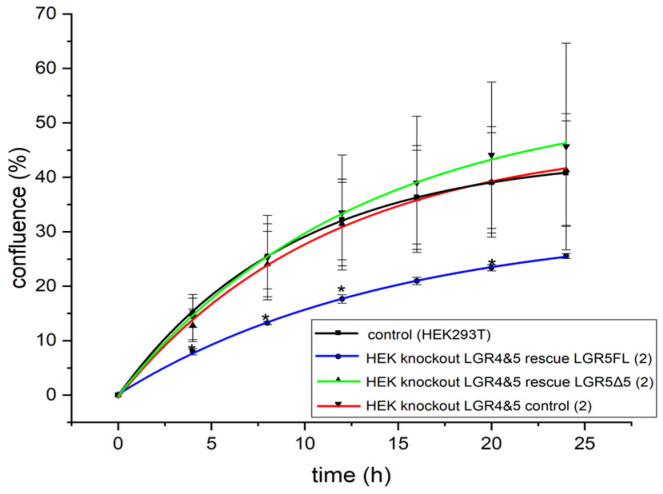
Migration of a HEK-LGR4/5 double-knockout clone with stable LGR5FL or LGR5Δ5 expression compared with the empty vector control cell line or unmodified HEK293T cell line via a scratch assay.

**Figure 8 ijms-25-13417-f008:**
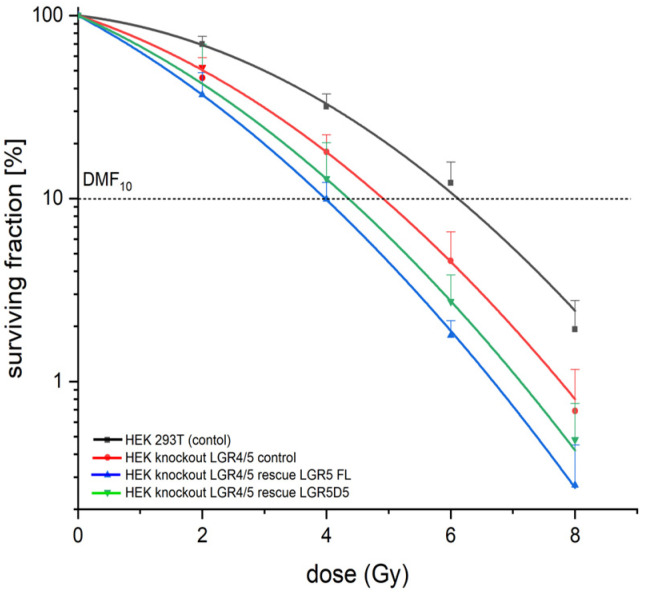
Radiosensitivity of the unmodified HEK293T cell line and the HEK-LGR4/5 double-knockout cell lines with stable LGR5FL or LGR5Δ5 expression or empty vector expression.

**Figure 9 ijms-25-13417-f009:**
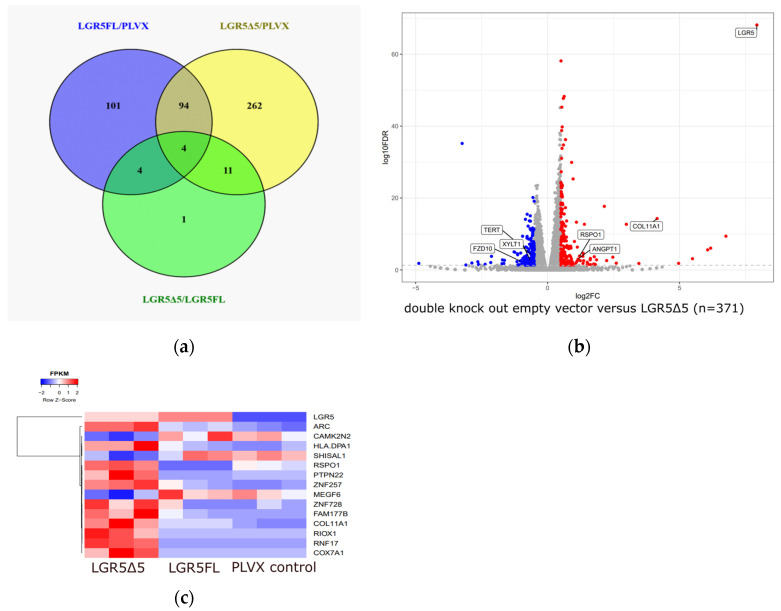
(**a**) Venn diagram of the HEK293T-LGR4/5 knockout cell lines (empty vector PLVX, LGR5FL, or LGR5Δ5 clones). (**b**) Volcano plot showing genes differentially expressed between the empty vector PLVX and LGR5Δ5 of the HEK293T-LGR4/5 knockout cell lines. (**c**) Heatmap of the expression of 15 genes related to LGR5Δ5 and LGR5FL overexpression (HEK293T-LGR4/5 knockout cell lines).

**Figure 10 ijms-25-13417-f010:**
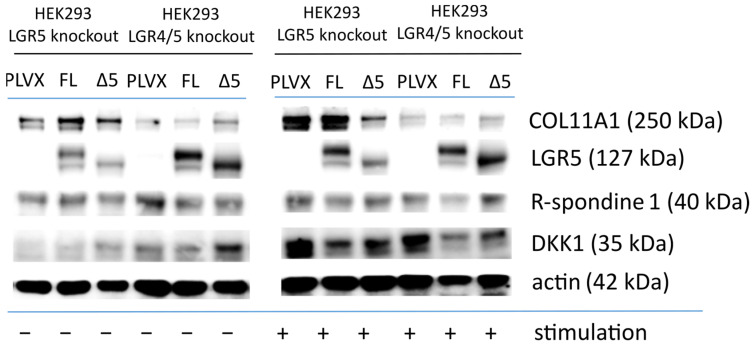
Western blot analysis of LGR5-regulated proteins in full-length LGR5 (FL)- and LGR5Δ5 (Δ5)-overexpressing HEK293T cells with single (LGR5) or double (LGR4/5) CRISPR Cas9 knockout with or without stimulation (Wnt3a and RSPO1) compared with control cells (PLVX) transfected with an empty vector. LGR5 and LGR5Δ5 were tagged with a YFP tag (27 kDa) for selection. ß actin was included as a loading control. Protein was isolated from adherent cells.

**Figure 11 ijms-25-13417-f011:**
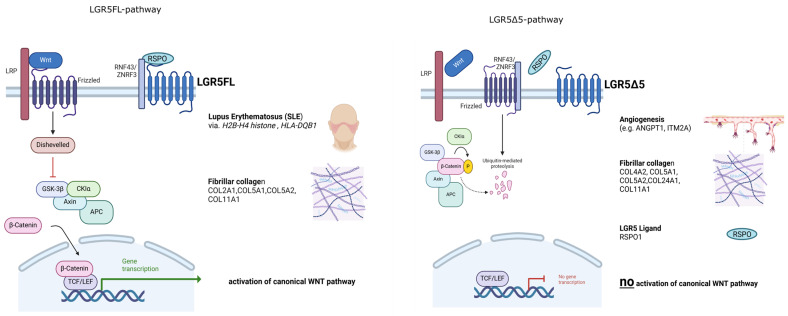
Postulated pathways associated with LGR5FL or LGR5Δ5 overexpression. LGR5FL activated the Wnt pathway, but LGR5Δ5 did not. LGR5FL overexpression is related to lupus erythematosus disease and fibrillar collagen regulation, whereas LGR5Δ5 overexpression is related to angiogenesis and fibrillar collagen regulation pathways. Our data also suggest that LGR5FL influences the level of genes associated with protein modification, transcription efficiency, and DNA accessibility adaptation in terms of epigenetic modifications. LGR5Δ5 seems to affect RSPO1 (the ligand of LGR5), an autoregulatory feedback loop activity of RSPO1. (Created with BioRender.com).

**Table 1 ijms-25-13417-t001:** Dose-modifying factor 10 (DMF10) values of the linear quadratic regression of relative plating efficiencies against the irradiation dose of all the HEK double-knockout clones with standard deviation (SD). DMF10 = DMF10 treatment/DMF10 control (HEK293T). *p* value: two-sided, paired *t* test for HEK293T cells. Biological triple test, two-sided paired *t* test. * represent a significant effect.

Cell Lines	DMF_10_	SD (+/−)	*p*-Value
HEK293T	1	-	-
HEK knockout LGR4/5 (empty vector)	1.272	0.1083	0.0490 (*)
HEK knockout LGR4/5 rescue LGR5FL	1.536	0.1168	0.0155 (*)
HEK knockout LGR4/5 rescue LGR5Δ5	1.462	0.125	0.0235 (*)

## Data Availability

Data is contained within the article and Appendix A.

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
