# Peer review of "Functional and Biological Characterization of the LGR5Δ5 Splice Variant in HEK293T Cells"

_ijms, 2024, doi:10.3390/ijms252413417_

Round 1
Reviewer 1 Report
Comments and Suggestions for Authors
Kappler et al has provided an interesting manuscript, titled “Functional and biological characterization of the LGR5Δ5 splice variant in HEK293T cells”. They demonstrate that LGR5 has an isoform, LGR5Δ5 which their group previously associated with patient prognosis and metastasis in OSCC. Their study evaluates the impact of LGR5Δ5 on Wnt signaling by utilizing CRISPR/Cas9 knockout of LGR5 and LGR4, with overexpression of LGR5Δ5 and LGR5FL rescue. They found that only LGR5FL could rescue migration defects, but not LGR5Δ5. Bulk-Seq analysis demonstrates an increase of RSPO1 and regulation of cytoskeletal, ECM and angiogenesis genes in LGR5Δ5 overexpression of LGR4/5 KO. Bulk-Seq analysis demonstrates regulation of collagens and histone proteins in LGR5FL. However, before their manuscript can be accepted for publication several revisions must occur.
Revisions:
1. Figure 2: While the authors show a time dependent internalization of LGR5FL and LGR5Δ5 after 20 minutes, what happens after 60 minutes or 120 minutes? Is there perhaps a delay in the return to baseline or prolonged internalization period for LGR5Δ5 mutants? Please quantify this figure as well and provide statistical analysis.
2. Line 129, the ending statement, per the data should be “The results revealed that the endocytosis of LGR5FL and LGR5Δ5 was via clathrin-dependent internalization.”
3. Figure 6: Authors should look into functional western blot readouts of Wnt signaling, like phosphor-LRP6.
4. Figure 10: Authors should run an IP reaction to see if there is differential binding of LGR5Δ5 and LGR5FL to RSPO1, given their schematic of Fig. 1b.
Reviewer 2 Report
Comments and Suggestions for Authors
The research design of this work is very well organized and the experiments are sufficient to prove the authors hypothesis. However, the way the results were described and discussed made the manuscript long and difficult to understand, in addition to taking some of the focus away from the LGR5Δ5 variant, which is in the title of the manuscript.
In my opinion, the authors should reformulate the results section, focusing the description of the results in the LGR5Δ5 and not in LGR5FL. Moreover, the description of the gene expression analysis in the results is too extensive and difficult to compare among the different groups. The authors should simplify the description.
The conclusions section is completely focusing on LGR5FL, which contrasts with the title of the manuscript.
Round 2
Reviewer 1 Report
Comments and Suggestions for Authors
No further comments, thank you for your thoughtful responses.